# “Diagnosis in the Prime of Your Life”: Facilitator Perspectives on Adapting the Living Well with Dementia (LivDem) Post-Diagnostic Course for Younger Adults

**DOI:** 10.3390/bs15060794

**Published:** 2025-06-09

**Authors:** Greta Wright, Natasha S. Woodstoke, Emily Dodd, Richard Cheston

**Affiliations:** 1School of Social Sciences, University of the West of England, Bristol BS16 1QY, UK; greta.wright140@gmail.com (G.W.); richard.cheston@uwe.ac.uk (R.C.); 2School of Health and Social Wellbeing, University of the West of England, Bristol BS16 1QY, UK; emily.dodd3@uwe.ac.uk

**Keywords:** Alzheimer’s disease: psychosocial intervention, social support, young-onset dementia

## Abstract

The Living Well with Dementia (LivDem) group intervention aims to support people to adjust following a diagnosis of dementia and is delivered across the UK and abroad. However, LivDem was designed for older people with dementia and may not address the needs of younger adults. This study aimed to identify the perspectives of LivDem facilitators on adapting the LivDem course for younger adults. Data was collected as part of an online facilitator survey and included questions requiring either ordinal or free-text responses. Responses from fifteen facilitators were analysed using descriptive statistics and Reflexive Thematic Analysis. The former indicated that participants believed that LivDem could be beneficial for younger adults and were in favour of it being adapted. Qualitative analysis generated two main themes, the first of which (‘The domino effect’: Unique Challenges for Younger Adults) had two subthemes: ‘Life and opportunities stripped away’ and ‘Impacting on everyone’. Theme 2, ‘Good to be with peers’: The Importance of Age-Appropriate Support, also had two subthemes: Groups ‘full of old people’ and Groups ‘specifically for younger people’. These findings reinforce the argument for creating age-appropriate services for people with young-onset dementia and will inform an adapted version of LivDem that provides age-appropriate support.

## 1. Introduction

Across the UK an estimated 944,000 individuals are living with dementia ([1]), with a further 55 million people affected worldwide ([47]). Whilst dementia has many different causes, all forms of dementia are characterised by a progressive loss of cognitive functioning which interferes with a person’s daily life and activities. Consequently, people living with dementia experience a range of difficulties from its onset, when any cognitive decline may be barely noticeable, to its most severe stage, when the person comes to depend almost completely on others. While there are pharmacological and non-pharmacological treatments for dementia, it cannot, as yet, be cured.

Dementia, then, represents an existential threat to a person’s identity, creating profound challenges for those who are affected ([15]). One such challenge is that of adjusting to the psychological and social changes inherent in dementia ([8]). Post-diagnostic social and emotional support can play an important role in supporting adjustment by validating experiences, promoting coping strategies, and helping people living with dementia to develop their understanding of the condition and how it impacts on them ([2]).

### 1.1. Young-Onset Dementia

The term ‘young onset dementia’ refers to those people who are aged under 65 when they receive a dementia diagnosis ([12]). When compared to the impact of dementia in older age, young-onset dementia causes a range of unique social and psychological challenges ([25]). An estimated 70,800 individuals are living with young-onset dementia in the UK ([1]). Younger adults are more likely to have a rarer form of dementia, such as frontotemporal dementia, to report significantly higher psychological and physical distress, and to have caring responsibilities themselves ([49]). Furthermore, the loss of meaningful activity, such as employment, is associated with diminished self-worth and disempowerment, all of which can lead to people with young-onset dementia being reluctant to disclose their diagnosis and avoiding social contact ([25]). Many younger people report intense fears about the future and even, at times, suicidal thoughts following their dementia diagnosis ([9]).

While services that focus specifically on the needs of younger people living with dementia are beginning to emerge, these are still relatively limited. Consequently, the unique needs of younger adults living with dementia often continue to be over-looked ([38]). While younger people living with dementia may well benefit from some of the opportunities for support aimed at older people, such as cognitive behavioural therapy for depression and anxiety ([22]) and dementia cafés ([26]; [20]), there is a clear demand for post-diagnostic support that has been specifically tailored for this population ([33]).

### 1.2. The LivDem Post-Diagnostic Course

The Living Well with Dementia (LivDem) course is a psychologically informed group intervention run by trained facilitators that is aimed at supporting people living with dementia to adjust to, accept, and come to terms with a dementia diagnosis. While LivDem is primarily a psychoeducational intervention, it incorporates some elements from a psychotherapeutic framework and is designed to be delivered by non-psychologists ([17]). It is an eight-week programme intended for around six to eight group members, all of whom will have received a diagnosis of dementia.

LivDem takes a deliberately slow pace to discussing dementia, with the initial sessions focusing on the symptoms experienced by group participants (e.g., short-term memory loss) but without explicitly associating these with a specific diagnosis. As the course progresses the group is encouraged to discuss their emotional responses to these cognitive changes, including how they cope with feelings such as embarrassment, anxiety, and depression. Only in weeks five and six do facilitators directly address issues around diagnosis and prognosis, for instance, by asking participants who they have told about the diagnosis. The final weeks address practical issues around living well. This design means that group participants are not forced to directly address the label of dementia in early sessions when this might be too threatening and uncomfortable for them. Instead, this is left until later sessions when they have had a chance to build a relationship with the other participants and a group has formed ([18]; [14]). The group format enables people who have been diagnosed to realise that they are not alone and allows them to learn from others going through similar experiences ([16]).

### 1.3. Adapting LivDem for Younger Adults

As the LivDem workbook for facilitators ([18]) was designed to support groups for older people living with dementia, it offers relatively little guidance about how the needs of younger people can be meaningfully addressed. Yet, younger adults living with dementia and their families often look for opportunities to develop their awareness of their condition and to have emotional support ([21]) and greater contact with others who share their diagnosis ([25]) in order to adjust to the diagnosis and the associated changes ([19]). While it might be possible for LivDem to meet these needs, post-diagnostic support needs to be age-appropriate if it is to provide positive experiences ([25]). This research therefore aimed to consider how the current LivDem model could be adapted to address the unique challenges that younger adults living with dementia experience.

### 1.4. Research Questions

This study set out to describe the perspectives of LivDem facilitators on adapting LivDem for younger adults by investigating two questions:
What are the psychosocial challenges faced by younger adults following a dementia diagnosis?How can LivDem best be adapted to meet the needs of younger people following diagnosis?

## 2. Materials and Methods

### 2.1. Design

The LivDem 2024 online survey gathered both quantitative and qualitative data via a semi-structured, anonymous questionnaire that was open between 7 December 2023 and 12 March 2024. The target participants worked within healthcare settings in the UK and abroad in both statutory and voluntary sectors. We specifically recruited LivDem facilitators as participants as they both had practical experience of delivering LivDem courses and were likely to have experience of delivering these and other services to younger people living with dementia. This research was approved by the University of the West of England Psychology Ethics Committee on 20 November 2023 (ref: LB2023006).

### 2.2. Participants

Around two hundred LivDem facilitators, previously trained by the LivDem team at the University of the West of England, were approached to take part through an advertisement in a newsletter and advertising on social media. Thirty-two respondents completed all or part of the survey, and after participants who had not progressed past the demographic questions were removed, fifteen participants were included in the analysis. The mean age of the participants was 39.13 (*SD* = 12.86), with fourteen out of the fifteen participants (93%) being women and thirteen identifying as white British (87%). All the names of the participants have been altered (see Table 1).

### 2.3. Procedure

Following the completion of the survey, participants received a debrief that included information on the research aims and how they could withdraw their data, as well as the contact information of the researchers. The raw data was exported to an Excel spreadsheet on a secure university server and accessed only by GW and RC. This data will be held until six months after the final paper resulting from this research is published and for a maximum of seven years.

### 2.4. Analysis

Descriptive analysis of the demographic and statistical data was carried out using JAMOVI^®^ 2.3. Qualitative responses were analysed using Reflexive Thematic Analysis, which is a qualitative research method that emphasises the role of the researcher’s self-awareness and critical reflection throughout the analytical process, acknowledging that researchers’ perspectives and experiences shape the way in which they identify themes and shape their interpretation of the data. Reflexive Thematic Analysis is a flexible technique that can be applied to a range of qualitative data, including material generated by online surveys ([7]). Our analysis followed [7]’s ([7]) six-phase guide, and while the analysis is described in chronological order, in actuality these processes were dynamic and non-linear ([45]). During phase 1 (familiarisation), the data was read and re-read, with the researchers noting their initial impressions within a reflexive journal. After preliminary immersion into the data, phase 2 (generating initial codes) was conducted using open coding, in which any interesting data was identified line-by-line to represent meaning as communicated by the participants ([6]). In phase three (generating themes), codes were written onto Post-it notes^®^, arranged by similarity, and copied into a table, moving the analysis from semantic description to interpretation by exploring broader meanings ([5]). Central organising concepts ([6]) were created to capture these clusters of codes and underpin the initial themes identified. During phase 4 (reviewing themes), a preliminary thematic mapping ([30]) was used to determine how the themes overlapped and interacted with one another, resulting in some themes being combined, removed, or split. The revised themes and thematic map were applied to the codes and extracts to ensure they reflected the entire data set ([10]) and to facilitate the transition to phase 5 (defining and naming themes), which attempted to capture the essence of findings. During phase 6 (producing the report), extracts from the coded data were used to illustrate aspects of the themes and explore their meanings, using the previous literature to further develop the findings ([10]).

### 2.5. Reflexive Statement

All of the authors have a background in psychology, which includes training in qualitative research methods: GW has an undergraduate degree in psychology and co-created the survey as part of that degree, during which she was supervised by RC; NW and RC are both clinical psychologists; and ED is a researcher with a psychology and public health background. Along with Ann Marshall, RC is the co-author of the LivDem manual for groups, whilst ED has worked on developing the LivDem approach for a number of years. NW has joined the LivDem team more recently. As an analytical team we attempted to be mindful of the balance between RC and ED’s commitment to the LivDem approach and a position of open enquiry as to whether the model needed to be adapted for those with younger-onset dementia and, if so, what such an adaptation might look like. Collectively, we have been influenced by the social model of disability ([36]), which suggests that we are “not disabled by our impairments but by barriers we face in society” ([36]).

## 3. Results

### 3.1. Quantitative Results

A forced-response five-point Likert scale ([31]) indicated that almost all the participants were either somewhat likely (44%) or extremely likely (44%) to recommend the LivDem course to younger adults living with dementia. None of the respondents reported that they would be unlikely to recommend it to younger adults. However, the proportion of referrals for people with young-onset dementia to the services in which the respondents worked was low (see Figure 1). Indeed, ten of the fourteen participants reported receiving either no referrals for younger people or less than 5% of the total number. One respondent worked for a “specific service for younger people (under 65s)” and thus reported that 100% of their referrals were for younger adults.

We were aware that there can be challenges in delivering face-to-face services in areas where there are relatively few people with younger-onset dementia and where it may be practically difficult to attend a group, for example, in rural areas. Therefore, we asked respondents about the online delivery of the course as a potentially more practical option for meeting the needs of younger adults. Facilitators were split on this, with 7 out of 14 (50%) of the respondents reporting that a hybrid model could be used and 6 (43%) stating that only in-person sessions should be offered.

### 3.2. Reflexive Thematic Analysis

The Reflexive Thematic Analysis of the survey responses generated two main themes (see Table 2) associated with adapting the LivDem post-diagnostic course for younger adults living with dementia. Theme one (‘The domino effect’: Unique Challenges for Younger Adults), details some of the ways dementia impacts daily life for those with an early onset of the condition. Theme two, (‘Good to be with peers’: The Importance of Age-Appropriate Support), offers insights into what content should be included within the LivDem course to tailor it to the experiences of younger adults.

#### 3.2.1. Theme 1: “The Domino Effect”: Unique Challenges for Younger Adults

Theme one explores younger adults’ concerns relating to employment, driving, familial responsibilities, and relationships. Together, the two subthemes of “Life and Opportunities Stripped Away” and “Impacting on Everyone” suggest that facilitators are highly aware that the timing of a young-onset diagnosis can heighten the sense of loss across many aspects of daily life.

##### Subtheme 1: ‘Life and Opportunities Stripped Away’

Ten participants emphasised the multiple challenges and losses many younger adults face when they are diagnosed. For facilitators, the intensity of these losses appeared to be related to the relative stage of their life cycle that younger adults were at compared to older adults. For example, potential participants are still likely to be working, and so a diagnosis of dementia may have financial and practical implications:

Antony: *“Particularly concerned about employment and financial worries if they can no longer work (or drive) following diagnosis.”*

Natalie: *“Acknowledging the trauma of losing your job/fear of losing your job.”*

Natalie’s use of the term trauma suggested that a loss of employment evokes negative emotions and provides additional stressors that may impact an individual’s ability to cope with a dementia diagnosis. Natalie went further in suggesting that financial worries were not the only implications of losing one’s job, with an individual’s identity and self-esteem also being affected:

Natalie: *“Lack of role and sense of purpose (in addition to reduced finances) if they have to stop work.”*

The mention of the lack of a role and sense of purpose indicates that employment is a large source of identity and structure for younger adults. As with working, the participants described concerns associated with a loss of driving as deeper than financial worries, specifically noting the impact on individuals’ independence:

Ophelia: *“Driving is also a key issue—where individuals may already [have] been advised to stop driving, which obviously impacts significantly on independence and access to occupations.”*

Participants suggested that the accumulation of these losses for younger adults is particularly intense due to the unexpected timing of their diagnosis:

Kiera: *“A loss of self can often be more problematic for this age group as they are still in the thick of developing their various roles & positions societally. There’s often more anger & frustration at having life & opportunities stripped away.”*

Natalie: *“Having this diagnosis in the “prime” of your life…”*

Antony: *“Shock at the diagnosis as it is ‘too early’ for this problem, it is something that happens to much older people.”*

These three powerful extracts suggest that younger adults do not expect the diagnosis and are more shocked and intensely impacted by the associated losses.

##### Subtheme 2: ‘Impacting on Everyone’: Changes in Roles and Relationships

Ten respondents spoke about their concerns that a dementia diagnosis extended beyond the self and the impact that this could have on the family unit as roles and relationships shifted:

Kiera: *“The decision about whether or not to tell the people around them carries even more significance, because of the additional impact it has on others … the knock-on effects on loved ones can be carried with guilt & shame.”*

Natalie: *“Impact of changes of role within the family unit, having child, grandchild and/or elderly parent responsibilities. The domino effect of one element impacting on everyone (emotionally, practically and financially).”*

Whilst Natalie highlights the multiple people to whom an individual may have responsibilities and thus the wide-ranging impact of a dementia diagnosis, Kiera refers to the guilt and shame that can be felt by individuals who observe the effects of their dementia on their loved ones. The use of the word “carried” in relation to these emotions conveys a weight to this experience, and she observes that these emotions may impact on whether people share their diagnosis, even with those to whom they are closest.

Some participants specifically mentioned the concerns of those with young-onset dementia about the impact on their children and grandchildren:

Debbie: *“Concerns about not being able to support their children or to get to know their grandchildren.”*

Antony: *“Uneasy that their adult children may be thrust into the role of caring for them so soon compared to someone diagnosed in their late 70s or 80s.”*

Some facilitators, then, were aware that as the families of many people with young-onset dementia would be relatively young, potential participants in a LivDem group would have to navigate how their relationship with their children might need to change. This highlights a clear difference between the challenges that younger and older adults with dementia face when adjusting to a dementia diagnosis. Thus, Antony’s use of ‘uneasy’ indicates an element of guilt that could be felt by younger adults relating to the impact of their dementia on young children. Antony also noted that sexual intimacy and individuals’ relationship with their partner may be a more dominant concern for younger adults living with dementia:

Antony: *“Particular dynamics which may be more prevalent in younger people’s minds might include the effect on possible sexual intimacy and their relationship with partner/spouse.”*

While changes in their levels and patterns of sexual intimacy may impact on people living with dementia of all ages, it may be that this is of especial concern for younger adults. This has a number of implications for the current model of LivDem. For instance, facilitators may need to develop ways to enable group participants to discuss these sensitive issues together. Additionally, families may need to be further integrated into the model to provide the opportunity for these relationship challenges to be explored and addressed ([46]).

#### 3.2.2. Theme Two: ‘Good to Be with Peers’: The Importance of Age-Appropriate Support

For many younger people, attending a LivDem course typically involves being part of a group of perhaps six to eight other people, most or all of whom are considerably older than they are. The second theme concerns how this difference in the ages of those attending post-diagnostic support groups such as LivDem could impact on the success and appropriateness of the sessions. There was a strong sense in participants’ accounts that bringing together people of a similar age was important and that this could potentially be achieved through online groups. Theme two is embodied by two subthemes: groups ‘Full of Old People’; and groups ‘Specifically for Younger People’.

##### Subtheme: Groups ‘Full of Old People’

This subtheme was present in the accounts of eight participants and captured the difficulties that can arise when a younger adult living with dementia joins a predominantly older support group, including concerns that their needs may not be addressed:

Eva: *“For most part those with YOD have not gained a lot from joining a group of older adults. They tend to assign themselves as ‘carers to the elderly participants’ rather than being able to focus on their own struggles.”*

Natalie: *“Not age appropriate. Very much in the minority. Needs were over-shadowed by the majority attending (older people).”*

Eva and Natalie both expressed their concern that as younger members are often in the minority within a group, their experiences are typically not at the forefront of group discussions. If younger adults were not able to “focus on their own struggles” (Eva) or experienced their needs being “overshadowed” (Natalie), then this would limit their opportunities to use the course to address the challenges they faced in living with dementia. Furthermore, if younger individuals perceived that the group was for older people they might feel out of place, perhaps limiting the connections they could make with other group members.

Other participants expanded on the way in which younger people with dementia were at risk of feeling out of place within predominantly older groups and highlighted instances of isolation and fear. Layla described the groups as being full of old people” and how younger attendees “found it frightening (seeing people in more advanced stages of dementia)”. Ophelia identified similar issues:

Ophelia: *“…some people report feeling alienated if other group members are much older than them or content is not suited to their age experiences.”*

The phrase “full of old people” (Layla) suggests that younger attendees may feel excluded from and different to the rest of the group and thus experience a sense of “alienation” (Ophelia). This sense of alienation is likely to negatively affect both their experience of the group and their adjustment to their diagnosis, which LivDem aims to support.

##### Subtheme: Groups ‘Specifically for Younger People’

Subtheme two, present in the account of 14 participants, captures the potential benefits of bringing similar-aged people with dementia together:

Natalie: *“Good to be with peers/people their own age or of a similar age. Shared experience.”*

Jane: *“If there were enough people living with young onset dementia referred then I think it would be of great benefit to run a course specifically for this client group. Peer support is always the most beneficial outcome of the course cited by attendees.”*

Ophelia: *“One specifically for younger people and their families. They feed back the value of meeting with others who are in a similar situation (for family members too) and also learning about information and resources that can help.”*

Jane and Ophelia suggested that for younger people being with others of a similar age could be beneficial as it would enable individuals in similar situations to share their experiences and knowledge. Some participants felt that offering similar-aged groups was particularly important as younger adults may otherwise not have the opportunity to meet peers of their own age and could become isolated:

Debbie: *“It might be harder for people with younger onset dementia to adjust to their diagnosis because it might be more unexpected, and they could feel more isolated/disconnected from those around them due to their age and seeing that others of a similar age are not experiencing the same types of problems that they are.”*

Debbie indicated that individuals with young-onset dementia can become isolated from their peers, as those of a similar age rarely experience the same problems. Participants emphasised the desire to ensure similar-aged groups but felt constrained by the fact that relatively few younger adults were referred to their services: 

Antony: *“Although we do not have enough referrals for people with young onset dementia, we do try to stream referrals so that there will be at least one other person in the group who is a similar age (i.e., under 65).”*

This indicates that the relative scarcity of people with young-onset dementia within these services may impede the ability of facilitators to establish a LivDem group solely for younger people. One option to address this issue would be to offer LivDem online, which would enable it to be offered over a larger geographical area so the low referral rates would be less of an issue. This flexibility might also offer benefits for those participants who are still working:

Charlotte: *“Younger people may still be in work and so do not have the time to attend sessions…hence why having the online sessions would be a useful option too. It also means you could reach people regardless of distance. Younger people are usually better gripped with technology to attend online sessions compared to older adults.”*

While Charlotte described the benefits of offering online courses, other participants highlighted the challenges inherent in delivering LivDem online—for instance, online sessions may make it much harder to meet the emotional needs of course members:

Kiera: *“In person groups are more appropriate when trying to build trust & open up potentially weighty conversations. It’s easier to focus when you’re not online. You can comfort someone properly f2f if they become upset or overwhelmed.”*

For Kiera supportive relationships were better fostered in-person than online. One potential solution that was suggested was to have a mix of in-person and online sessions that would permit more flexibility whilst still ensuring a comfortable environment for individuals to express difficult emotions:

Fiona: *“People prefer different ways so offering both may help with engagement. Sometimes having an online option is more convenient and easier to attend. Sometime people prefer the face-to-face interaction, and it is easier to interact and build relationships.”*

## 4. Discussion

This mixed-methods study had two aims: to investigate the perspectives of LivDem facilitators on the challenges faced by younger adults living with dementia and to understand how the LivDem course might be adapted to respond to some of these challenges. Importantly the quantitative findings show that facilitators would recommend LivDem to younger adults, thus supporting the idea of adapting the course.

Theme one (‘The domino effect’: Unique Challenges for Younger Adults) described a range of ways in which a diagnosis can impact younger people and how this might be taken into consideration when adapting LivDem. The participants indicated that the time of diagnosis causes compounded stress due to it being while individuals are at working age and have more caring responsibilities. Previous research has similarly found that dementia-related losses and changes occur at a time of high financial, familial, and occupational responsibilities ([13]), intensifying stress for younger adults.

The participants highlighted the financial and emotional concerns associated with the loss of jobs and driving abilities, which can impede the ability of younger people to live well with dementia. This supports previous research reporting that a loss of income and unplanned retirement can induce additional strain ([40]) and impact on younger adults’ sense of self and purpose ([39]). Furthermore, the loss of the ability to drive can reduce independence ([43]) and cause younger adults to perceive themselves as a burden, worsening depressive symptoms ([40]). Whilst both older and younger adults report grief over an accumulation of losses ([13]), younger adults may feel this more intensely due to these changes happening at an earlier age than expected ([27]).

The first theme also demonstrated concerns about the impact of dementia on family roles and relationships, specifically parent–child relationships, and sexual and emotional intimacy with partners. [27] ([27]) have proposed that when younger people with dementia are unable to perform their expected family roles, such as parenting duties, then stress and conflict can occur, suggesting that family worries can impact on a person’s ability to live well with dementia. Furthermore, after diagnosis the quality of relationships often changes as caring needs become more pronounced and the focus shifts from intimacy towards dependency ([19]). Declines in relationship quality, where intimacy and sexuality are a lower priority, are associated with increased frustration, tension, and isolation ([28]). This may impact on a person’s ability to live well with dementia as shared love and commitment may be integral aspects of their shared sense of couplehood ([28]; [37]). Both the partners and children of those living with dementia have highlighted that communication about the diagnosis is important for adaptive coping ([3]; [42]), suggesting that encouraging conversations with family members can facilitate families living as well as possible.

Theme 1 highlighted that younger adults may have a wide range of concerns that could potentially impede their ability to accept and adjust to their diagnosis. Unique challenges such as the loss of their job, the need to stop driving, and changes to family roles and relationships may be at the forefront of younger adults’ minds, provoking additional fear, guilt, and stress. As LivDem aims to facilitate discussions of diagnosis and its implications and reduce the fear around dementia, these could be important topics to consider incorporating into a course tailored for younger adults living with dementia.

Theme two (‘Good to be with peers’: The Importance of Age-Appropriate Support), offered insight into some of the difficulties in providing effective support for younger people. First, it highlighted how the predominantly older demographics of typical LivDem participants may impede access for younger adults and negatively impact on those who attend. Younger adults have previously reported viewing dementia support groups negatively ([43]), as they mainly cater for older adults ([38]).

Alongside the concerns raised about predominantly older groups, the results suggest that a LivDem course specifically for younger adults could potentially be beneficial. Prior research has also found that similar-aged groups are more inclusive, as they enable participants to receive support from those who understand their unique situation ([41]), thereby fostering empowerment ([38]) and allowing for advice regarding specific daily challenges to be shared ([23]). However, the rarity of young-onset dementia and lack of referrals to these services reported here reiterates [24]’s ([24]) concern that the limited flow of younger adults through these services may restrict the number of specific groups for young people that post-diagnostic services can run.

One possible solution to this barrier may be the delivery of online sessions—however, respondents in our study differed in how helpful they thought this might be. Online sessions were seen as ways of overcoming difficulties for potential participants in getting to sessions and would allow for more flexibility, which younger adults may need due to their caring commitments and employment ([41]). However, participants felt that either in-person or a mixture of online/in-person sessions would be the most appropriate. This echoes the work of [23] ([23]), who found that individuals accessing online dementia peer support sessions enjoyed attending from the comfort of their own homes but missed in-person sessions. However, in contrast to these concerns LivDem was successfully adapted online in Italy, where the group continued to meet a year after the course ended ([48]), suggesting that good group relations can indeed be fostered online.

### 4.1. Implications

While the relatively small sample size of facilitators means that the further validation of the results would be helpful, this research could nevertheless inform adaptions to the LivDem course so that it is more accessible and impactful for younger adults living with dementia. The reduced access to post-diagnostic support for younger adults living with dementia is widespread ([35]) and is not just limited to LivDem; therefore, this research may offer insight for other post-diagnostic courses seeking to tailor sessions for younger adults living with dementia.

This research also reinforces the need to move away from pharmacological approaches toward emphasising wellbeing in dementia care ([29]). Furthermore, acknowledging and respecting younger adults’ unique experiences enables person-centred support to be based on individual preferences and priorities ([4]). Adapting post-diagnostic interventions for younger individuals is vital as they embed coping strategies early on in dementia progression ([2]), a time when the experience of loss is especially prominent for younger adults ([44]).

### 4.2. Strengths and Limitations

While LivDem facilitators are ideally placed to be able to comment on potential adaptations of the intervention, we only succeeded in recruiting a relatively low number of respondents, with just fifteen facilitators completing the survey. This total was fewer than the number who participated in two similar online surveys of LivDem facilitators: [16] ([16]) reported on twenty-three participants while [17] ([17]) recruited twenty-eight. It is not clear why the rate in this study was lower, but as 32 facilitators completed the initial demographic questions but only 15 went onto to complete the survey, it is possible that some experienced a technical difficulty while accessing the survey and were unable to continue. At the same time the use of qualitative analysis to complement the quantitative questions enabled us to potentially identify good practice ([10]). Importantly, our research did not attempt to engage with younger people living with dementia and their families, a vital step towards establishing a holistic understanding of the needs of this client group.

All but one of the participants within this study were white British females, and all of them were working in roles that required a degree. While this perhaps reflects the overreliance of psychological sciences on WEIRD (Western, Educated, Industrialised, Rich, and Democratic) samples ([34]), the homogenous nature of LivDem facilitators also raises other issues. Although LivDem facilitators are likely to be largely women, estimates of the sex ratio in people with young-onset dementia vary, but most have found that there are more men than women. A French study ([11]) of 18,466 people with young-onset dementia found that 54% were men and 46% women, whilst a UK study found a 3- to 4.7-fold greater prevalence in males than in females ([32]), although this sex distribution has not been supported by all studies. It may be that in some cases at least, younger men with dementia, particularly if they are not white or have not been to university, may be reluctant to engage with professionals who are largely white, female graduates. This disparity in backgrounds may also impact on the health professional–patient relationship and the ability of some facilitators to empathise fully or engage with the experiences of and challenges faced by all members of the group.

### 4.3. Recommendations

Future research should explore younger adults’ own perspectives on adapting the LivDem course. Younger adults could offer insight and reflections about their experiences of adjusting following a diagnosis and thus shape an adapted LivDem intervention to ensure it meets the needs of those it aims to help.

## 5. Conclusions

This study explored some of the unique and sometimes overwhelming challenges faced by younger adults who are diagnosed with dementia. In line with previous research, the results support the need for post-diagnostic interventions to be tailored to the specific needs of this group. A failure to focus on the unique needs of younger adults for support means that many will continue to struggle unnecessarily to adjust and cope following a dementia diagnosis.

## Figures and Tables

**Figure 1 behavsci-15-00794-f001:**
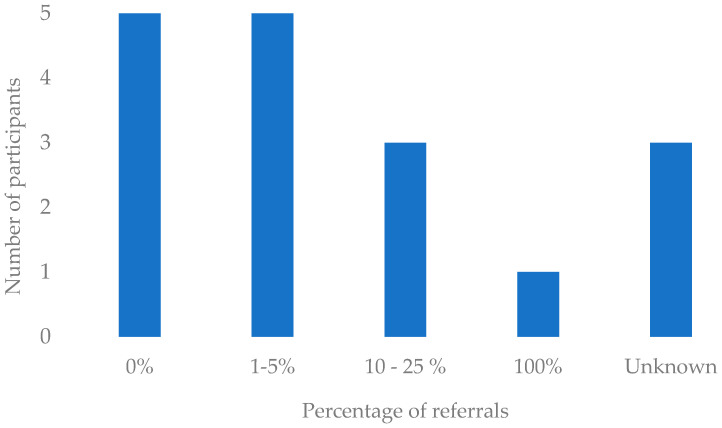
Estimated proportion of young-onset referrals to the services in which the facilitators worked.

**Table 1 behavsci-15-00794-t001:** Description of participant characteristics.

Pseudonym	Age	Gender	Ethnicity	Country of Practice	Professional Background	Number of LivDem Courses Delivered or Supervised
Antony	55	Male	White British	UK	Support worker	8
Beth	29	Female	White	UK	Psychologist	1
Charlotte	26	Female	White British	UK	Assistant psychologist	0
Debbie	23	Female	White British	UK	Assistant psychologist	0
Eva	59	Female	White British	UK	Clinical psychologist	5
Fiona	25	Female	White British	UK	Assistant psychologist	1
Georgia	23	Female	White British	UK	Assistant psychologist	3
Heather	26	Female	British	UK	Assistant psychologist	4
Isabel	42	Female	White British	UK	Support worker	2
Jane	49	Female	White British	UK	Occupational therapist	6
Kiera	36	Female	White British	UK	Therapist	4
Layla	46	Female	White British	UK	Mental health nurse	0
Mary	53	Female	White British	UK	Therapist	9
Natalie	49	Female	White British	UK	Occupational therapist	0
Ophelia	46	Female	White British	UK	Occupational therapist	4

Qualtrics^®^ software (Qualtrics, Provo, UT, USA, release date 13/12/2023) was used to create the questionnaire for the survey. This consisted of thirteen questions, five of which required ordinal responses and eight free-text answers (see Appendix A).

**Table 2 behavsci-15-00794-t002:** Themes and subthemes.

Themes	Subthemes	Examples of Codes	Number of Participants Coded for in Subtheme (Pseudonyms)
‘The domino effect’: Unique Challenges for Younger Adults	‘Life and opportunities stripped away’	Feelings of loss Financial worries Losing purpose and routine Loss of ability to drive impacts independence Loss of self Diagnosis in ‘prime’ of life	10 (Antony, Beth, Debbie, Eva, Isabel, Jane, Layla, Mary, Natalie, Ophelia)
	‘Impacting on everyone’: changes in roles and relationships	Performing parenting duties Caring responsibilities Effects on loved ones Impact of guilt and shame due to diagnosis	10 (Antony, Beth, Debbie, Eva, Isabel, Jane, Layla, Mary, Natalie, Ophelia)
‘Good to be with peers’: The Importance of Age Appropriate Support	Groups ‘Full of Old People’	Not able to focus on own struggles Needs overshadowed by those of older group members Full of old people Stigma of dementia as older persons’ disease	8 (Beth, Eva, Heather, Isabel, Jane, Layla, Natalie, Ophelia)
	Groups ‘specifically for younger people’	Valuable meeting others in similar situation More isolated and disconnected from age group Peer support hugely beneficial Online course accessible for people without transport Course needs more flexibility	14 (Antony, Beth, Charlotte, Debbie, Eva, Fiona, Georgia, Isabel, Jane, Keira, Layla, Mary, Natalie, Ophelia)

## Data Availability

The data is available from the lead author on reasonable request.

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
