# Peer review of "“Diagnosis in the Prime of Your Life”: Facilitator Perspectives on Adapting the Living Well with Dementia (LivDem) Post-Diagnostic Course for Younger Adults"

_behavsci, 2025, doi:10.3390/bs15060794_

Round 1
Reviewer 1 Report
Comments and Suggestions for Authors
Thank you so much for the opportunity to review this paper and I really enjoyed reading it as it is very well written and highly relevant to the field of behavioural sciences. This article presents the findings which reinforce the argument for creating age-appropriate services for people with young onset dementia. it will be of great value to anyone interested and/or involved in the development, delivery and receipt of person-centred services.
I have only a small number of comments which I set out below.
Keywords:- Alzheimer's disease plus it might be useful to add 'young onset dementia'
l 33 - is 'initial' needed here?
l 36 - can the cause of a disease be cured - is that the right term (it might be)?
l 101 - should it be 'aimed' and there is a word missing - possibly 'describe' or 'identify'
l 107 - I am wondering about the word 'unique' here. This includes, later on. aspects which are not unique to young onset dementia. Is it the combination of challenges that is unique?
l 131 - Qualtrics needs registered trademark symbol
l 140 - JAMOVI needs registered trademark symbol
l 140 - the sentence starting 'Qualitative responses' - should say what reflexive analysis is. Then a new sentence starting 'This approach - but the rest of the sentence would need re-writing as it doesn't quite make sense.
l149 - post it notes needs registered trademark symbol
l 156 - should it be 'a thematic map' or 'thematic mapping'?
l 163 - 'all authors have a background in psychology' - does this link to the points l 168 on? If so how?
l 45 - needs a little more elaboration as to how
l 465 - there is a bracket missing. I feel the sentence needs re-writing it is not that clear as to meaning/easy to read.
I hope these suggestions are helpful and thank you once again for this opportunity.
Author Response
Reviewer 1
Thank you so much for the opportunity to review this paper and I really enjoyed reading it as it is very well written and highly relevant to the field of behavioural sciences. This article presents the findings which reinforce the argument for creating age-appropriate services for people with young onset dementia. it will be of great value to anyone interested and/or involved in the development, delivery and receipt of person-centred services.
I have only a small number of comments which I set out below.
Keywords:- Alzheimer's disease plus it might be useful to add 'young onset dementia'
we add 'young onset dementia' to the keywords
l 33 - is 'initial' needed here?
the word 'initial' was not needed and so removed this
l 36 - can the cause of a disease be cured - is that the right term (it might be)?
While there are pharmacological and non-pharmacological treatments for dementia, it cannot, as yet, be cured
l 101 - should it be 'aimed' and there is a word missing - possibly 'describe' or 'identify'
removed ‘aimed’ replaced ‘set out’
l 107 - I am wondering about the word 'unique' here. This includes, later on. aspects which are not unique to young onset dementia. Is it the combination of challenges that is unique? –
removed ‘unique’
“This study set out to describe the perspectives of LivDem facilitators on adapting LivDem for younger adults, by investigating two questions”
l 131 - Qualtrics needs registered trademark symbol
l 140 - JAMOVI needs registered trademark symbol
All three words (Qualtrics, JAMOVI and post it) have had a registered trademark symbol added after them
l 140 - the sentence starting 'Qualitative responses' - should say what reflexive analysis is. Then a new sentence starting 'This approach - but the rest of the sentence would need re-writing as it doesn't quite make sense.
Qualitative responses were analysed using Reflexive Thematic Analysis which is a qualitative research method that emphasizes the role of the researcher's self-awareness and critical reflection throughout the analytical process, acknowledging that the researchers' perspectives and experiences shape the way in which they identify themes and shapes their interpretation of the data. Reflexive Thematic Analysis is a flexible technique that can be applied to a range of qualitative data including material generated by online surveys (Braun and Clarke, 2019).
l149 - post it notes needs registered trademark symbol
All three words (Qualtrics, JAMOVI and post it) have had a registered trademark symbol added after them
l 156 - should it be 'a thematic map' or 'thematic mapping'?
mapping
l 163 - 'all authors have a background in psychology' - does this link to the points l 168 on? If so how?
We have clarified this link by adding the phrase “which includes training in qualitative research methods”
l 45 - needs a little more elaboration as to how
as for clarification here
l 465 - there is a bracket missing. I feel the sentence needs re-writing it is not that clear as to meaning/easy to read.
Rewritten the paragarph
I hope these suggestions are helpful and thank you once again for this opportunity.
Reviewer 2 Report
Comments and Suggestions for Authors
The objectives are relevant and timely, as there is growing recognition of the distinct needs of people with young-onset dementia. I have a few comments that the authors shall justify with reason.
- The sample size is only fifteen facilitators, while qualitative exploration with the help of this sample restricts generalizability, especially geographical distribution. As you mentioned in the limitation, however, I suggest adding explanations in the Implications section also.
- Instead of only feedback from the facilitator, why don't you take direct input from younger adults with dementia and their caregivers for a better, holistic understanding?
- Please also justify why the dropout rate was high, and if you used any strategies to prevent it?
- It is more useful if you provide the professional background of the participants like years of experience, working setting etc.
- The nature of questions (theme, number,) as mentioned in semi structured questionnaire id not described.
- One brief note is needed on how the confidentiality of data security was maintained.
Author Response
Reviewer 2
The objectives are relevant and timely, as there is growing recognition of the distinct needs of people with young-onset dementia. I have a few comments that the authors shall justify with reason.
- The sample size is only fifteen facilitators, while qualitative exploration with the help of this sample restricts generalizability, especially geographical distribution. As you mentioned in the limitation, however, I suggest adding explanations in the Implications section also.
The first sentence of the implications section has been amended to include a reference to the relatively small sample size. We have added additional clarification that this is lower than in previous surveys, and we speculate that this may have been due to the high drop out rate (over 30 people completed the demographic questions), which in turn might have been attributable to a technical problem preventing further access to the survey - Instead of only feedback from the facilitator, why don't you take direct input from younger adults with dementia and their caregivers for a better, holistic understanding?
Added a reference to this in the limitations section – “Importantly, our research did not attempt to engage with younger people living with dementia and their families – a vital step towards establishing a holistic understanding of the needs of this client group.” - Please also justify why the dropout rate was high, and if you used any strategies to prevent it?
As above - we have commented on this and suspect that this may have been due to a technical problem preventing access – for instance if participants tried to compete it from an on-compatible device - It is more useful if you provide the professional background of the participants like years of experience, working setting etc.
We unfortunately did not collect any more information than we report – we have changed ‘job title’ to professional background’ in table 1 - The nature of questions (theme, number,) as mentioned in semi structured questionnaire id not described.
Added an appendix - One brief note is needed on how the confidentiality of data security was maintained.
The raw data were exported to an excel spreadsheet on a secure, university server, and accessed only by GW and RC. This data will be held until six months after the final paper from the research is published and for a maximum of seven years.